# Influence of North Pacific Decadal Variability on the Western Canadian Arctic over the past 700 years

**François Lapointe[1,2], Pierre Francus[1,2], Scott F. Lamoureux[3], Mathias Vuille[4], Jean-Philippe Jenny[1,5], Raymond S. Bradley[6], Charly Massa[7]**

[1]Centre - Eau Terre Environnement, Institut National de la Recherche Scientifique Université du Québec, Québec, G1K 9A9, Canada

[2]GEOTOP Research Center, Montréal (Qc), H3C 3P8, Canada

[3]Department of Geography and Planning, Queen's University, Kingston, ON K7L 3N6, Canada

10 [4]Department of Atmospheric and Environmental Sciences, University at Albany, Albany, New York 12222, United States.

[5]Max-Planck-Institute for Biogeochemistry, 10, 07745 Jena, Germany

[6]Northeast Climate Science Center, and Climate System Research Center, Department of Geosciences, University of Massachusetts, Amherst, 01003, United States.

[7]Department of Geography, University of Hawaii at Mānoa, Honolulu, HI 96822, United States.

15 Correspondence to: François Lapointe (francois.lapointe@ete.inrs.ca)

**Abstract**

Understanding how internal climate variability influences arctic regions is required to better forecast future global climate variations. This paper investigates an annually laminated (varved) record from the western Canadian Arctic and finds that the varves are negatively correlated with both the instrumental Pacific Decadal Oscillation (PDO) during the past century and also with reconstructed PDO over the past 700 years, suggesting drier Arctic conditions during high PDO phases, and vice-versa. These results are in agreement with known regional teleconnections whereby the PDO is negatively and positively correlated with summer precipitation and mean sea level pressure, respectively. This pattern is also evident during the positive phase of the North Pacific Index (NPI) in autumn. Reduced sea-ice cover during summer-autumn is observed in the region during PDO- (NPI+) and is associated with low-level southerly winds that originate from the northernmost Pacific across the Bering Strait and can reach as far as the western Canadian Arctic. These climate anomalies are associated with the PDO- (NPI+) phase and are key factors in enhancing evaporation and subsequent precipitation in this region of the Arctic. Collectively, the sedimentary evidence suggests that North Pacific climate variability has been a persistent regulator of the regional climate in the western Canadian Arctic. As projected sea-ice loss will contribute to enhanced future warming in the Arctic, future negative phases of the PDO (or NPI+) will likely act to amplify this positive feedback.

## 1 Introduction

In the North Pacific region, the Pacific Decadal Oscillation (PDO) is the major mode of multi-decadal climate variability (Mantua et al., 1997). The PDO can be described as a long-lived El Niño/Southern Oscillation (ENSO)-like pattern of Pacific sea surface temperature (SST) variability (Allan et al., 1996; Zhang et al., 1997), or as a low-frequency residual of ENSO variability on multi-decadal time scales (Newman et al., 2003). During the warm (positive) PDO phase (PDO+), regions of southeast Alaska, the southwestern US and Mexico generally have increased winter precipitation, whereas drier conditions are observed in southern British Columbia and the Pacific Northwest US. During PDO- conditions

are essentially reversed (Mantua and Hare 2002). To date, little is known, however, about the influence of the PDO on the climate of the Canadian Arctic. Indeed, the impacts of large-scale mode of climate variability in this region have not been documented because of the lack of 1) reliable meteorological datasets, which generally don't extend prior to 1950, and 2) annually-resolved climate archives.

In the recent years, several varved records have been established in the Arctic (at Cape Bounty: Cuven et al. (2011); Lapointe et al. (2012), at South Sawtooth Lake: Francus et al. (2008), at Lake C2: Douglas et al. (1996), at Murray Lake: Besonen et al. (2008) and Lower Murray Lake: Cook et al. (2009)) in order to investigate past climate variations. Amongst them, the Cape Bounty record is most probably the best documented because it has been supported by climate, hydrological, and limnological research at the Cape Bounty Arctic Watershed Observatory since 2003. The annual nature of this

sedimentary record, its duration (700 years), and the above-average quality of its chronology opens the opportunity to investigate (1) correlations with instrumental records, (2) cyclities of this record by time-series analysis, (3) teleconnections with major climate indices, and (4) the long-term influence of the climate mode of variability on the western Canadian Arctic.

## 2 Materials and methods

2.1 Study site

        Cape Bounty East Lake (hereafter CBEL, 5 m asl, Fig. 1 black asterisk) is located on southern Melville Island in the Canadian Western High Arctic (74° 53' N, 109° 32'W). CBEL is a small (1.5 km$^2$) and relatively deep (32 m) monomictic freshwater lake. The lake has ice cover for 10-11 months of the year and has one primary river inflow. CBEL has been monitored since 2003 as part of comprehensive hydrological and limnological studies that revealed the nature of sediment

delivery and deposition in this setting (Cockburn and Lamoureux, 2008; Lamoureux and Lafrenière, 2009; Lewis et al., 2012). Fluvial input to the lake occurs mainly during June and July during spring snowmelt and also due to major rainfall events generally later in the summer season (Dugan et al., 2009; Lapointe et al., 2012; Lewis et al., 2012). Previous studies (Cuven et al., 2011; Lapointe et al., 2012) demonstrated the presence of clastic varves in the lake and documented the past hydroclimatic variability using the physical and geochemical properties of the varve sequence. Finally, seismic profiles of

the lake bottom revealed that the coring site used in Lapointe et al. (2012) and Cuven et al. (2011) was located away from mass movement deposits, therefore well suited for paleoclimatic investigations (Normandeau et al., 2016a, Normandeau et al., 2016b).

## 2.2 Observational climate data

To understand the recent relationship between the Western Canadian Arctic climate and the PDO, a one-point correlation map was calculated using the Pearson's correlation. These were prepared using the Climate Explorer tool that is managed by the Royal Netherlands Meteorological Institute (Trouet and Van Oldenborgh, 2013; Van Oldenborgh and Burgers, 2005). Precipitation, sea-level pressure, temperature and sea-ice anomalies were obtained from the ERA-Interim reanalysis (Dee et al., 2011), a dataset that provides robust observations of mean temperature and precipitation in the Canadian Arctic (Lindsay et al., 2014; Rapaić et al., 2015). For zonal and meridional wind, the NCEP-NCAR (Kalnay et al., 1996) which cover the period 1950-2016 was used. The PDO as defined in Mantua (1997) is derived as the leading principal component of monthly SST anomalies in the North Pacific Ocean, poleward of 20°N. A second PDO index was constructed by regressing the Extended Reconstructed Sea Surface Temperature (ERSSTv4) (Huang et al., 2015) temperature anomalies against the Mantua PDO index during the period of overlap. This resulted in a PDO regression map for North Pacific ERSST anomalies. This index closely resembles the Mantua PDO index. The NPI is described as the area-weighted sea-level pressure over the region 30°N-65°N, 160°E-140°W (Trenberth and Hurrell, 1994). Finally, the Arctic Oscillation Index, representing the leading Empirical Orthogonal Function (EOF) of monthly mean 1000 hPa geopotential height anomalies over 20°-90° N latitude (Thompson and Wallace 1998) was used. Finally, the Mould Bay weather station record, located 320 km northwest of CBEL, was extracted from: http://climate.weather.gc.ca/historical_data/search_historic_data_e.html.

## 2.3 Chronological control

The methods used to count varves rely on both visual examination of thin sections and the use of ~ 7000 microscopic images (1024 X 768 μm) obtained using a scanning electron microscope in backscatter mode. This technique allows for the reliable identification of thin varves (< 0.4 mm), thus decreasing the chances of missing thin varves (Ojala et al., 2012). The chronology of the recent part of the record was also confirmed by radiometric dating ($^{137}$Cs and $^{210}$Pb) (Cuven et al., 2011). Counts were made by two different users and yielded very similar results in the upper part (above 167 cm), in which the first 925 varves are present (1075 CE). The error between the two counts is estimated to be less than 1.2% (Lapointe et al., 2012), a very good number compared to other similar records (Ojala et al., 2012). Overall, the counts were very consistent since 244 CE implying that the varves from CBEL are well-defined and unambiguous (Lapointe et al., 2012). Only three coarse layers, dated 1971 CE (Lapointe et al. 2012), 1446 CE (Fig. S4) and 1300 CE (Fig. S6), are found in the 1750-year long sequence. These are the sole discernible features that have likely caused minor erosion in the sedimentary record from 1300-2000 CE (Figs. S4, S6). Moreover, CT-scans of the core record did not reveal any unconformities. Finally a recent acoustic survey revealed that the coring site was devoid of mass movement deposits (Normandeau et al., 2016). In brief, all these features are suggesting that the CBEL sedimentary record is minimally affected by erosion (Cuven et al., 2011; Lapointe et al., 2012).

## 2.4 Proxy data

Varve thickness and grain-size data (Lapointe et al., 2012), available from the NOAA paleoclimate database, were linearly detrended. A Box-Cox transformation was then used to stabilize variance in the time series (note that the use of both no transformation or a log-transformation of the time series yielded similar results). The data were normalized to allow for a comparison with other time series. Three PDO reconstructions (D'Arrigo et al., 2001; Gedalof and Smith, 2001; MacDonald and Case, 2005) were used for comparison with the CBEL record. Spectral analyses were carried out using REDFIT (Schulz and Mudelsee, 2002) and wavelet analyses were performed with the software R (Team, 2008) using the package biwavelet (Gouhier and Grinsted, 2012). For wavelet analysis the interval 244-2000 CE was analysed as the lake was fully isolated by glacioisostatic uplift from the ocean after 244 CE (Cuven et al., 2011; Lapointe et al., 2012).

# 3 Results

### 3.1 Instrumental teleconnections

Several key climate indices demonstrate the present-day influence of the PDO on the western Canadian Arctic. The correlation between the PDO index (Mantua et al., 1997) based on ERSSTv4 (Huang et al., 2015) and sea-ice cover (Dee et al., 2011) is positive during summer and autumn over the region (Figures 1a, S1). An anomalous surface high-pressure system develops in the vicinity of southern Melville Island from July to September (JAS) (Figure 1b) during positive PDO phases (PDO+). The PDO index is also inversely correlated with summer rainfall from the nearest continuous weather station, Mould Bay (Figure 1c), implying drier (wetter) conditions during the positive (negative) phase of the PDO (r = - 0.47, p < 0.0001). This suggests that PDO-related atmospheric circulation anomalies significantly affect the climate of this region (Fig. 1).

Another important teleconnection is revealed in the spatial correlation between PDO and mean sea level pressure (MSLP) during winter (Fig. 1d). The mid-to high-latitude manifestation of the PDO includes a wave train that is characterized by a deepening of the Aleutian Low and a high-pressure system to the northeast over the Canadian Arctic during PDO+, somewhat reminiscent of the Pacific - North America pattern PNA (Wallace and Gutzler, 1981), and most prominent during the positive phase of ENSO. Melville Island is located at the core of this teleconnection wave train, and is ideally located to sample extremes of the PDO as they are expressed as significant departures of MSLP during each phase (Fig. 1d). The existence of a persistent anomalous high-pressure system over this area during the PDO+ is indicative of drier than average conditions in the region, while negative MSLP anomalies during the negative PDO phase (PDO-) likely reflect the more frequent passage of low-pressure systems and the increased likelihood of precipitation (Fig. 1c).

The western Canadian Arctic is also strongly influenced by the North Pacific Index (NPI) during September-November (SON) (Fig. 2). The NPI is a more direct measure of the strength of the Aleutian Low (Trenberth and Hurrell,

1994) and has been shown to be part of the PDO North Pacific teleconnection (Schneider and Cornuelle, 2005). A weakened Aleutian Low (increased MSLP) is seen in the Pacific during times of positive NPI (NPI+), as is the case during PDO-. Meanwhile, an anomalous low-pressure system is observed over the western Canadian Arctic (Fig. 2a), consistent with an increased likelihood of precipitation (Fig. 2b). This is confirmed by the correlation between snow depth recorded at Mould Bay and the NPI (Fig. 2c).

### 3.2 Comparison with instrumental and paleo-PDO records

The sedimentary varve record gives support to these instrumental climate observations. Annual coarse grain-size (98[th] percentile) (Lapointe et al., 2012) is negatively correlated with the instrumental PDO (Mantua et al., 1997) during the last 100 years (no lag: $r = -0.26$, $p = 0.01$, maximum correlation at 1-year lag: $r = -0.31$, $p = 0.001$ and $r = -0.84$ using a 10-year low-pass filter, Fig. 3), suggesting thicker varves (deposits) during PDO-. A similar correlation is found between instrumental NPI (Trenberth and Hurrell 1994) and coarse grain-size at CBEL (Fig. S2 : $r = 0.30$, $p = 0.003$: no lag).

For the time interval 1300-1900 CE, a single 1.34 cm thin erosive bed is evident in the sedimentary record (Supplemental Text 1, Supplemental Fig. S4), making the comparison of the CBEL varve thickness (VT) with other paleo-PDOs acceptable. The three reconstructed PDOs (MacDonald and Case 2005, Gedalof and Smith 2001, D'Arrigo et al. 2001) show periods of high coherency, but there are periods of low consistency between them (Fig. 4a-c), as reported in the literature (Kipfmueller et al., 2012; Wise, 2015). These reconstructed PDOs are probably best interpreted as reflections of the PDO at their given study sites, explaining the lack of co-variability during certain periods. To better explain the variance in the paleo-PDO time series, a principal component analysis (PCA) was performed on the three reconstructed PDOs. The PC1 (Fig. 4d) explains 51% of the variability (loadings factors: 0.58 (D'Arrigo et al. 2001), 0.68 (Gedalof and Smith 2001) and 0.65 (MacDonald and Case 2005)) and its highest correlation with VT is achieved with an 18 year lag (Fig. S3: $r = -0.29$, $p < E-5$). Given the present-day teleconnection (Figs. 1-2) and the overall co-variability between the instrumental PDO and the CBEL record (Fig. 3), this lag is likely due to intrinsic errors of varve chronologies (Ojala et al., 2012) rather than a mechanistic phase shift. When applying a 5-year running-mean on the series the co-variability is striking ($r = -0.48$), especially from 1750-1900 ($r = -0.68$). From 1600-1900, annual correlation between Gedalof and Smith (2001) and the

CBEL record is significant (r = -0.21, p < 0.001). When a 5-year running mean is applied on the series, the coherence between both records is much stronger (Fig. 4b: r = -0.39). This is also the case when comparing the CBEL VT record to the D'Arrigo et al. (2001) PDO (Fig. 4c, annual correlation: r = -0.25, p < 0.001; 5 yr-running mean: r = -0.29). The correlation of the CBEL record to the PDO from MacDonald and Case for the period 1446-1900 is also significant (annual correlation: r = -0.24, p < E-7, 5 year-running mean: r = -0.39). For the period encompassing 1300-1446 CE, the records are significantly correlated with a 28 year-lag. This broader lag is likely related to erosion produced by a high-energy event (second largest layer of the record) dated at ~1446 CE (Fig. S4). When shifting our record back by 28 years, a high co-variability exists between both records (Figs. 4a, S5). The overall annual correlation with the MacDonald and Case (2005) index is slightly improved during the pre-industrial interval 1300-1845 CE (Figs. 4a: annual correlation: r = -0.27, p < E-10, r = -0.43 (5-year running-mean) and -0.69 (25-year low-pass filter, Fig. S5).

To obtain accurate confidence intervals for the linear correlation between the PDO records and CBEL, a nonparametric stationary bootstrap, using 1000 iterations, is used (Mudelsee, 2010). The optimal average block length is determined using the method described in Patton et al. (2009), which is well suited for autocorrelated time series. The correlation analysis performed on both raw and 5-year filtered data shows a large improvement of the significance levels with filtered data (Fig. S8) as well as stronger correlation coefficients (Table 1). Also, the use of filtered data provides narrower confidence intervals, that is, less uncertainty. The visible and statistically significant negative correlation between three independent PDO records and CBEL strongly support our assumption that the varve thickness at our site is influenced by the PDO.

**3.3 Spectral content of the 244-2000 CE period**

To further support the link between the Cape Bounty sequence and the PDO (NPI), spectral analysis of the entire VT record for the 244-2000 CE period found significant (> 99% confidence level (CL); Fig. 5a) spectral peaks at ~19-26 and at 62 years that are consistent with those found in the high-frequency (19-25 year) and also the lower frequency range (50-70 year) of the PDO (Chao et al., 2000; Latif and Barnett, 1996; Mantua and Hare, 2002; Minobe, 1997; Tourre et al., 2005). The 2-4 year-cycle in the VT could be linked to ENSO, which is characterized by high-frequency variability of 2-8 years

(Deser et al., 2010). Many significant sub-decadal periodicities at ~2-8 years are evident (Fig. 5b). These periodicities are particularly pronounced from 1450 to 2000 CE and 800 to 1200 CE. Over the last millennium, the 50-70 year oscillation has been persistent at Cape Bounty from ~1000 to 1550 CE and from ~1700 CE until recently (Fig. 5b). This is somewhat different from the PDO reconstruction from tree-rings (MacDonald and Case, 2005) in which the wavelet spectrum displays a persistent power band covering only the periods ~1350-1500 CE and 1800 CE until recently. Similar to MacDonald and Case (2005), CBEL reveals a weaker multi-decadal variability during the 17[th] century and the early part of the 18[th] century. However, in contrast to MacDonald and Case (2005), significant power located at 2-8 years remains relatively constant during most of the past millennium in CBEL and is particularly strong between ~850-1250 CE (Medieval Climate Anomaly, MCA), ~1450-1750 CE (coldest interval of the LIA), and recently (Fig. 5b). A ~60-year periodicity is also clearly discernible during 600-800 CE, a period also characterized by strong decadal and sub-decadal (2-7 year) cycles. Altogether, these relationships point toward a significant influence of the PDO on the western Canadian Arctic.

## 4 Possible mechanisms linking the CBEL record to the PDO

When the western Canadian Arctic is characterized by lower pressure system anomalies when the Aleutian Low is in a weakened state (increased SLP, NPI+, Fig. 2), it is plausible that the prevailing winds reaching the region originate from the northern Pacific. Indeed, a negative correlation between meridional windstress and the NPI during SON over the northernmost part of the Pacific and extending into the western Canadian Arctic (Fig. 6a) indicates prevailing northerly wind anomalies during the positive phase of the NPI. It has similarly been shown that PDO and Arctic Oscillation (AO) are useful determinants of precipitation characteristics during summer season in regions of Alaska (L'Heureux et al., 2004) and positive AO index has been linked to reduced sea-ice extent and increased atmospheric heat transport into the Arctic (Rigor et al., 2002; Zhang et al., 2003). The correlation between the AO and the meridional windstress anomalies (Fig. 6b) yields very similar pattern as the NPI (Fig. 6a). This is not too surprising, since these two climate indices are significantly correlated during SON (1900-2015: r = 0.45, p < 0.0001). Hence the two modes which may share in part the same signal might constructively interfere to strengthen northerly winds over the Arctic during AO+ and NPI+, converging with southerly

moisture-laden winds from the North Pacific over the western Canadian Arctic, thereby favoring precipitation in the region during autumn.

These meridional wind anomalies appear to persist during the cold season (Fig. S9), although they are not as pronounced over the western Canadian Arctic as in September-November (Fig. 6a). This is consistent with annual surface wind stress differences between PDO phases over the North Pacific (Zhang and Delworth, 2015) during the 20[th] century (Fig. 7). Indeed, sustained southerly wind anomalies are observed in the northernmost part of the Pacific during PDO- (induced by a weakened Aleutian Low, i.e. NPI+), north of the Kuroshio-Oyashio Extension, where warm SST anomalies are observed (Screen and Francis, 2016; Zhang and Delworth, 2015) (Fig. 7). These southerly winds extend from the northernmost Pacific (north of the weakened Aleutian Low) across the Bering Strait and can reach as far as the western Canadian Arctic, increasing heat and moisture transport to the latter region (Screen and Francis, 2016). Meanwhile, strong westerly winds dominate over the eastern Siberian shelf and converge with the southerly flow from the Pacific over the western High Arctic during PDO- (Kwon and Deser, 2007; Screen and Francis, 2016; Zhang and Delworth, 2015) (Fig. 7). Thus, the PDO phase has been shown to clearly influence the winter-mean atmospheric circulation in the North Pacific while its influence also extends into the Arctic (Screen and Francis 2016). Our analysis suggests that this PDO (NPI) influence might also be impacting regional climate during autumn.

Warmer summer temperatures during PDO- are also observed in large areas of the Arctic (Fig. S10a). This is most apparent in the western Canadian Arctic during NPI+ (Fig. S10b). It has been shown that PDO- (and NPI+) lead to lower tropospheric Arctic warming and sea-ice loss (Screen and Francis, 2016), and the combination of reduced sea-ice extent (Figs. 1, S1) and warmer surface temperature during PDO- (NPI+) (Fig. S10) likely allows for more evaporation to occur, while anomalous surface winds (Figs. 7, 8) increase moisture convergence in the region, thereby enhancing precipitation (Figs. 1c, 2b, c). Analyses by Francis et al. (2009) have shown that the Aleutian Low tends to be weaker following summers of reduced sea ice cover. A comparison between the CBEL record and instrumental sea-ice extent since 1979 (Cavalieri et al., 1996) (Fig. S11: $r = -0.52$, $p = 0.01$) suggests increased precipitation during times of low sea-ice extent. Winds during periods of a weakened Aleutian Low (Figs. 6, 7) and reduced sea-ice extent in the region, as seen during PDO- (Fig. 1a),

would likely be more effective at transporting moisture across the western Canadian Arctic (Fig. 2b). More importantly, Arctic sea-ice extent reached unprecedented low values in the latter half of the 20[th] century compared to the last 1450 years (Kinnard et al., 2011). This trend is similar to the coarse grain-size at CBEL, which increased substantially and reached unprecedented levels in the 20[th] century compared to the last 1750 years (Lapointe et al. 2012). All of these elements point to a causal mechanism, linking the NPI (PDO), sea-ice and precipitation in the western Canadian Arctic.

## 5 Conclusion

This study suggests a significant influence of the PDO (NPI) on the climate of the western Canadian Arctic, a region where instrumental data coverage is very sparse and the duration of available records is short. Spatial correlations using both instrumental and reanalysis data indicate a strong atmospheric teleconnection, likely responsible for the increase of precipitation during PDO- (NPI+). These results indicate the importance of large-scale teleconnections for Arctic climate and in particular, for precipitation variations in the Canadian High Arctic. An important finding from this study is the reduced sea-ice cover observed during PDO-, which is in agreement with simulations made from Screen and Francis (2016). The PDO – western Canadian Arctic relationship has persisted at least for the past ~700 years as revealed by the strong coherence between the CBEL varve record and multiple PDO reconstructions. Given the oscillatory nature of the PDO, there is some potential for improved constraint over decadal-scale climate prediction using the kind of sedimentary record shown here, which in turn could give insights into future sea-ice variability. In that sense, more high-resolution records with longer timescales from this region could be beneficial for future PDO projection.

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

# Acknowledgement

We wish to thank the Polar Continental Shelf Program for their field logistic support and NSERC grants to PF and SFL. FL is grateful to grants provided by the FRQNT and the W. Garfield Weston Foundation. We thank Geert Jan van Oldenborgh for advice with the use of the KNMI database. We also thank James Screen for constructive advice, and Byron Steinman and

5   Ze'ev Gedalof who provided information on PDOs datasets. FL would also like to thank Charly Massa, David Fortin and the Ouranos Consortium for constructive conversations. Paleo-data used in this study can be found on the NOAA server https://www.ncdc.noaa.gov/data-access/paleoclimatology-data/datasets

**Table 1: Correlation analysis for the varve thickness at Cape Bounty East Lake and different proxy records of PDO. r is Pearson's**
10  **correlation coefficient, and p is the probability that two uncorrelated time-series would exhibit a higher correlation. The percentile confidence intervals at 95%, calculated from 1000 nonparametric stationary bootstrap iterations, are indicated in brackets.**

| | $p$ | | $r$ | |
|---|---|---|---|---|
| Study | raw | filtered | raw | filtered |
| Gedalof and Smith (2001) | 0.05 [E-10; 0.646] | 0.008 [E-29; 0.053] | -0.19 [-0.35; -0.01] | -0.39 [-0.65; -0.19] |
| MacCase (2005) | E-04 [E-17; E-05] | E-10 [E-46; E-13] | -0.24 [-0.33; -0.15] | -0.42 [-0.55; -0.30] |
| D'Arrigo et al. (2001) | 0.29 [0.002; 0.92] | 0.08 [E-26; 0.81] | 0.01 [-0.04; 0.22] | -0.29 [-0.64; 0.08] |
| PC1 | 0.02 [E-11; 0.01] | E-03 [E-37; E-04] | -0.29 [-0.46; -0.10] | -0.53 [-0.74; -0.25] |

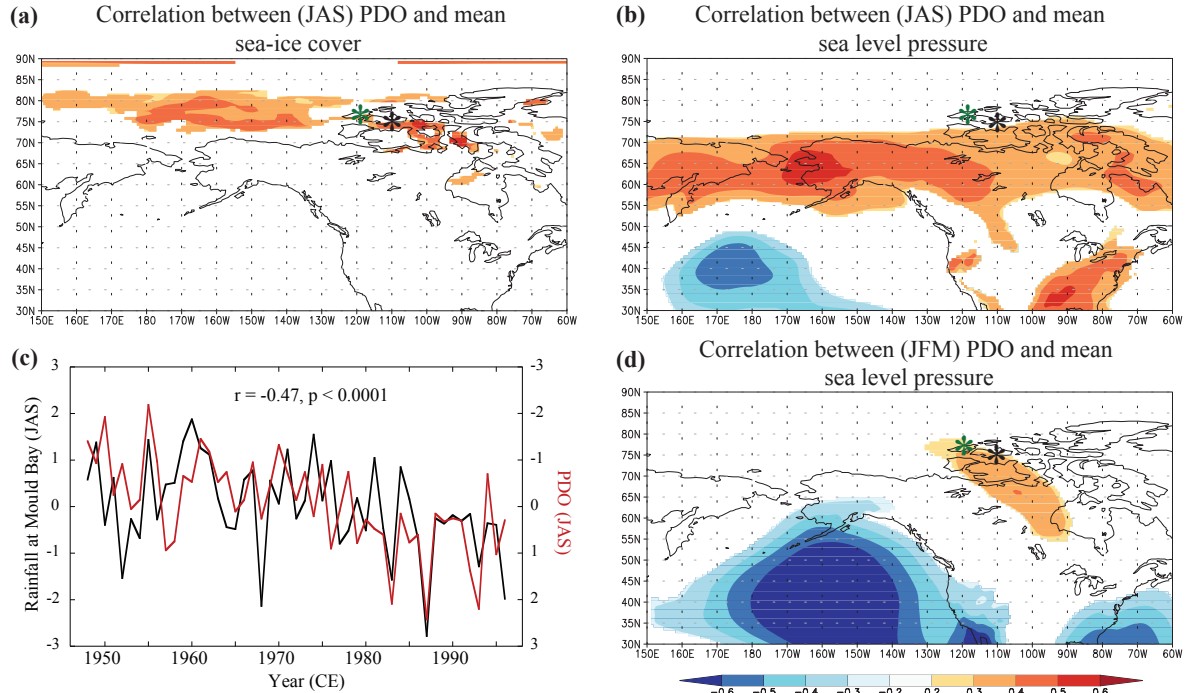

**Figure 1.** PDO modulation of Western Canadian Arctic climate. (a), Correlation between PDO (Huang et al., 2015) and sea-ice anomalies from ERA-Interim (Dee et al., 2011) for July-September during 1979-2016. (b), as in a) but for mean sea level pressure from ERA-Interim (Dee et al., 2011). (c), Comparison between the time series of rainfall at Mould Bay and PDO during July-September. (d), as in b) but for January-March (JFM). Black and green asterisks denote Cape Bounty and Mould Bay weather station, respectively. Note that Mould Bay weather station stopped operating in 1996.

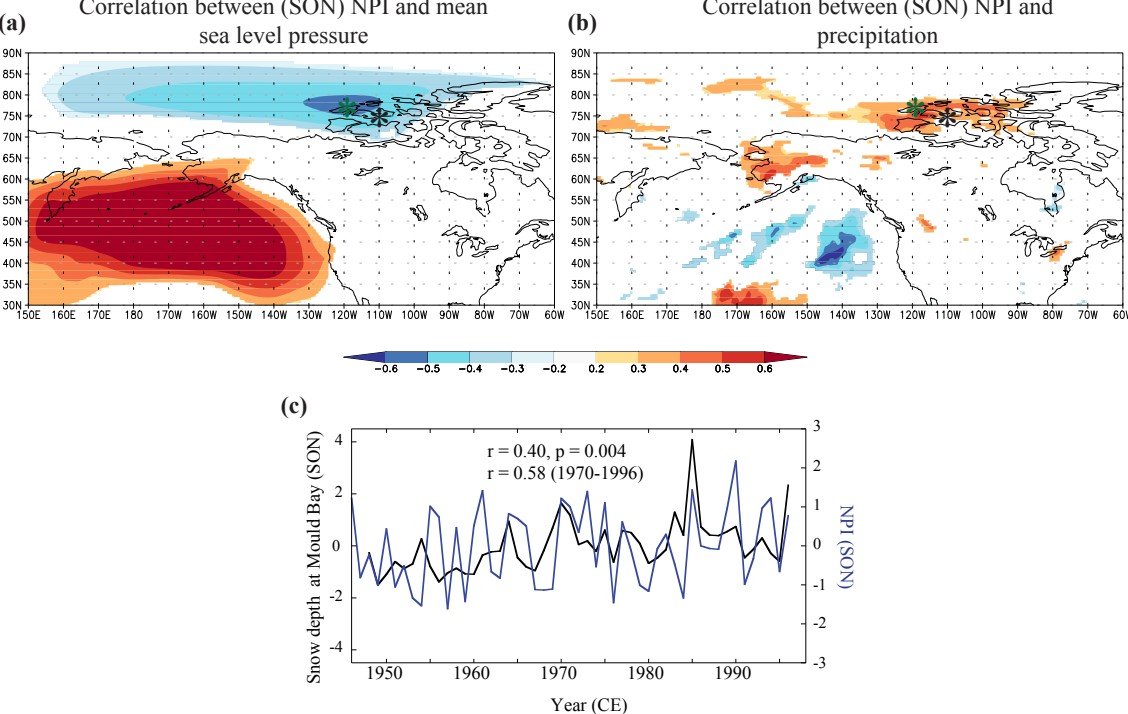

**Figure 2.** North Pacific Index (NPI) and precipitation during September-November. (a), Correlation between NPI (Trenberth and Hurrell, 1994) and mean sea level pressure from 1979-2015. (b), Same as (a), but for precipitation anomalies (Dee et al., 2011) correlated with NPI index. Black and green asterisks denote Cape Bounty and Mould Bay weather station, respectively. (c) Comparison between the time series at Mould Bay snow depth and NPI during September-November (Trenberth and Hurrell, 1994). Note that Mould Bay weather station stopped operating in 1996 CE.

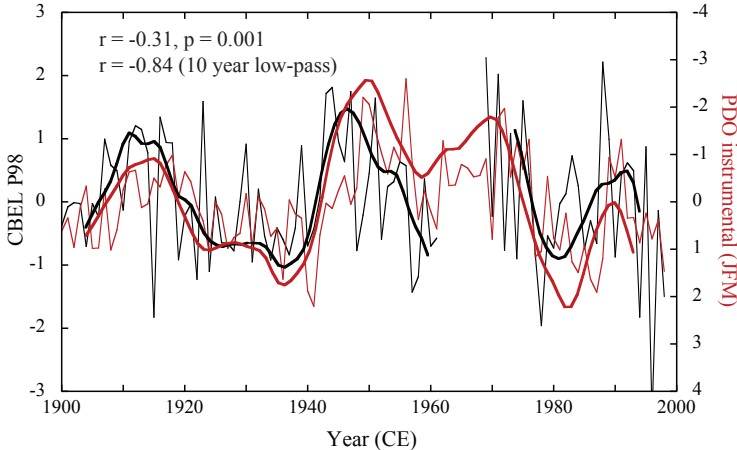

**Figure 3.** Instrumental PDO (NOAA) compared with grain size at Cape Bounty East Lake from 1900-2000. (best correlation is achieved when CBEL lags PDO by 1 year). Bold lines are 10-year low-pass filtered. Seven years were eroded by a large turbidite dated to 1971 CE (Lapointe et al. 2012).

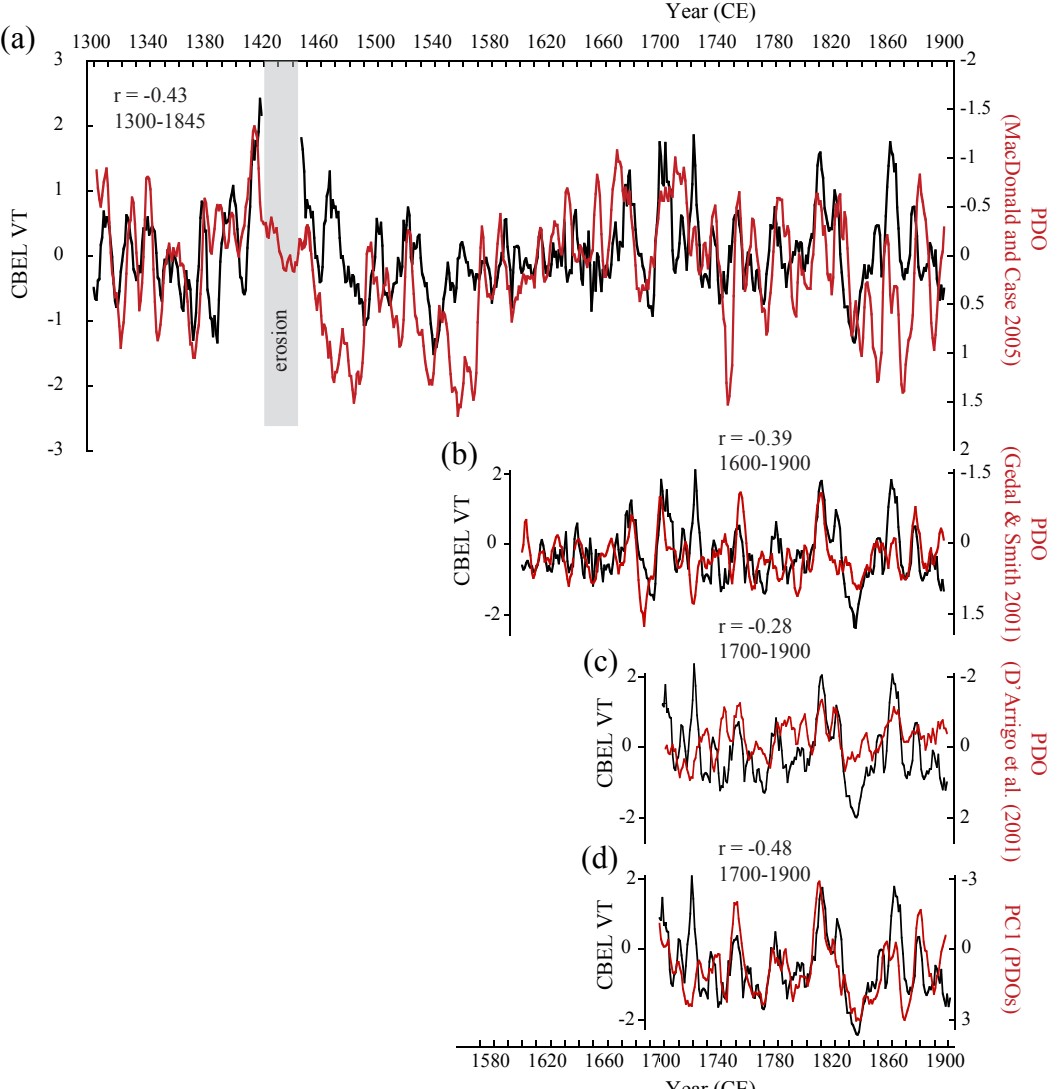

**Figure 4**. a), Comparison between normalized Cape Bounty East Lake varve thickness and normalized PDO from MacDonald and Case (2005) (VT is shifted 18 years earlier). b), Same as A) but for the PDO from Gedalof and Smith (2001). c), Same as a) and b) but using the PDO from D'Arrigo et al. (2001). d), Same as a), b) and c) but using the PC1 extracted from PCA analysis of the three PDOs. Time series are filtered by a 5-year running-mean.

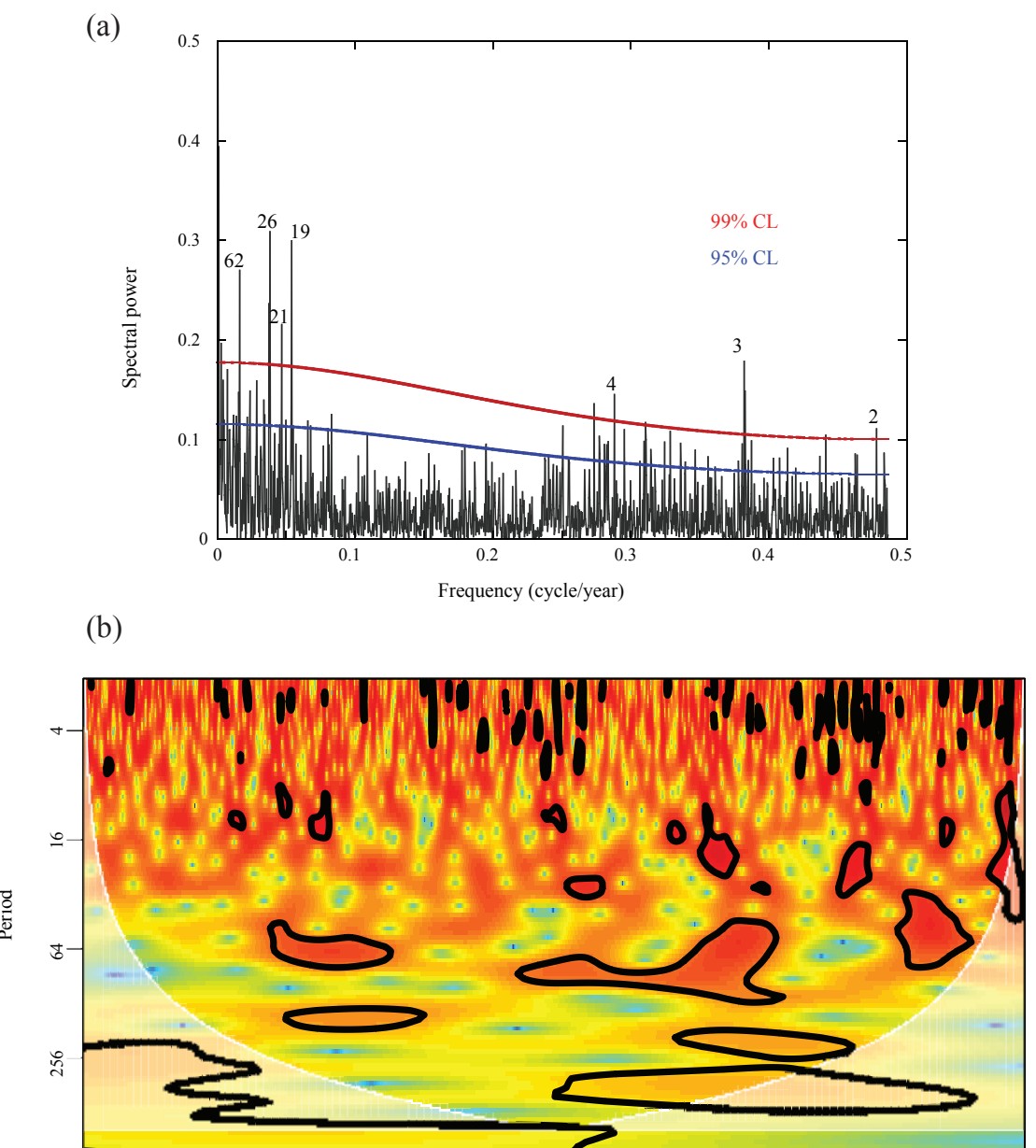

**Figure 5**. a), Spectral analysis of the varve thickness series. After Schulz and Mudelsee (Schulz and Mudelsee, 2002). b), Wavelet analysis: black boundaries show the 95% confidence level based on a red noise process. White shading represents the cone of influence where edge effects might be important.

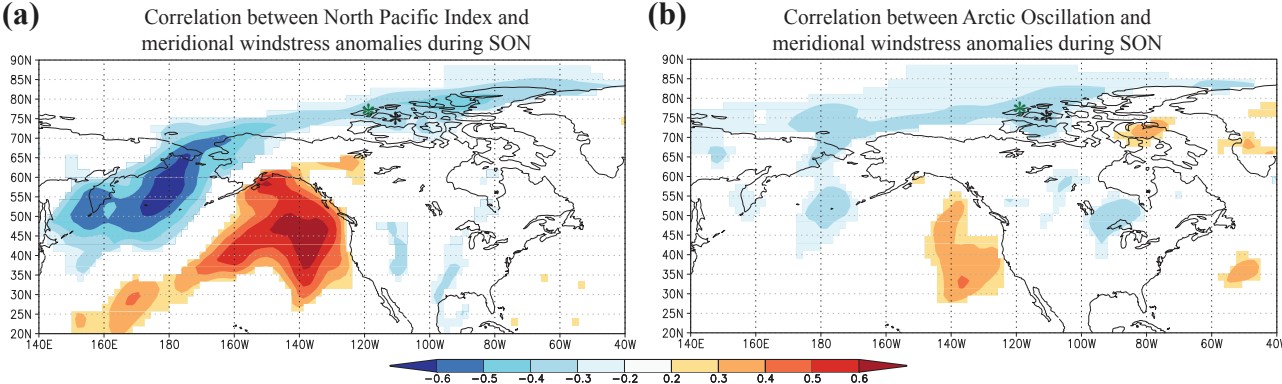

**(a)** Correlation between North Pacific Index and meridional windstress anomalies during SON

**(b)** Correlation between Arctic Oscillation and meridional windstress anomalies during SON

**Figure 6.** a), Correlation between NPI (Trenberth and Hurrell, 1994) and meridional windstress anomalies from 1950-2015. b), Same as a), but for correlation with Arctic Oscillation index (derived from NCEP/CPC). Note that the Era-Interim yields similar result (not shown). Black and green asterisks denote Cape Bounty and Mould Bay weather station, respectively.

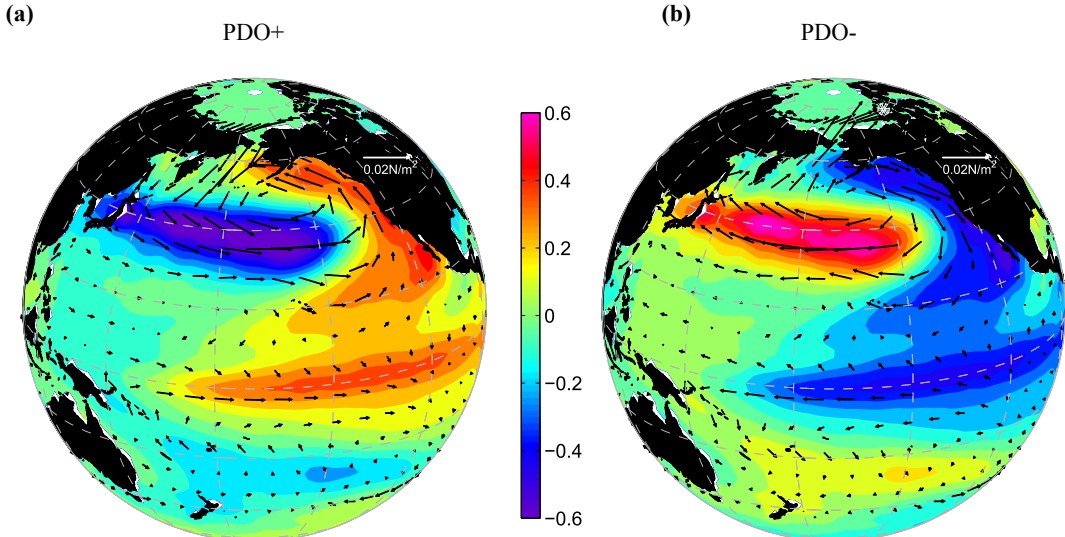

**(a)** PDO+

**(b)** PDO-

**Figure 7**. PDO modulation of winds and sea surface temperature in the Pacific. From Zhang and Delworth (Zhang and Delworth, 2015). Regression of SST (°C) and surface wind stress (N m$^2$) against the PDO index. Note the northward direction of the wind stress in the central northern part of the Pacific during the negative phase of the PDO (b). Winds from the Siberian shelf have an eastward direction and reach Melville Island during negative PDO. Reproduced with permission from the American Meteorological Society (AMS).

