# Peer review of "Influence of North Pacific Decadal Variability on the Western Canadian Arctic over the past 700 years"

_Climate of the Past, 2016_

## Referee Comment (RC1) · Anonymous Referee #1 · 1 Dec 2016

This is an interesting study that describes a 700-year annually-laminated varve record of past precipitation from sediment in Mould Bay in the western Canadian Arctic. It should be published with some revision.

General Comments: 1. I would like to have seen more discussion of dating accuracy in the main text, particularly given the discussion of sizeable errors of 18, 28 years in lags with the paleoclimatic comparisons. Also on how well the varves reflect lower frequency climatic information. 2. Perhaps some discussion of whether the PDO is the most significant influence to discuss here, rather than the Arctic Oscillation. 3. The tree-ring reconstructions of the PDO vary in part because of the different geographical representation of the sites used in each case. Another PDO reconstruction based

on tree rings is that of Biondi. As a result I believe that they are best interpreted as reflections of the PDO at their given study sites. 4. Are there perhaps other varve/paleo records in the vicinity of the varve site that might be more appropriate for comparison? 5. Good to note the issue of seasonality – trees reflect conditions during different seasons than the varves..also that the dating is more precise. 6. Some (mostly light) editing of English would benefit the manuscript.

Minor points: Abstract: References ok in abstract? Line 9, reword to note that negatively correlated with instrumental for past century, recons over past centuries to 700 years.. P. 3 line 7 ENSO references by Rob Allan, Hadley Centre relevant here p. 3 line 20 show varve site on map. How far from Mould Bay? p. 4 line 16 Mantua, 1997 p. 5 first paragraph: good to discuss errors in dating a bit here in main text.. p. 6 line 2: MSLP not mslp p. 6 line 22: inference not clear re erosive bed and how this relates to first part of sentence p. 6 line 26: what are the loadings of the three recons in PC1? p. 7 line 1 18 year lag: this seems like a rather large offset. 7 line 10: ditto 28 year lag..

---

## Referee Comment (RC2) · Anonymous Referee #2 · 22 Jan 2017

General comments

I think this is a potentially nice study on the influence of PDO on Western Canadian Arctic and on the mechanisms relating PDO and a varved record. However the main concern with the paper is that the authors do not clearly state their objectives and the links between the paper sections. At the end of the introduction we do not know if the paper is mostly a comparison between a varved record and PDO observations/reconstructions or if the authors want to study the PDO influence over the last century with correlations.

Specific comments

The abstract must be reworded. This is mostly a comparison between a varved record and PDO observations/reconstructions (P 2 L 6. "Here, sedimentological evidence from an annually laminated (varved) record highlights that North Pacific climate variability has been a persistent regulator of the regional climate in the western Canadian Arctic."). The conclusion of the abstract (P 2 line 15-20) says nothing on the results/implications of THIS paper. (PS Now that I have finished to read the paper I have partially changed my mind on this comment, however I see that the problem is that you do not clearly state your objectives and the methods you apply to reach them)

Introduction. The objectives of the study are not illustrated.

Section 2.2. You must describe here your data. Not at the end of the introduction which is the place for objectives.

Section 3.3. Do you think that the spectral analysis can also be influenced by the origin of the data (tree-rings, varved records) and not only by the modes? For example, you use a box-cox transformation to stabilize variance in your time series. What do you get in terms of spectral analysis if this transformation is not applied?

P 2 Line 10. "suggesting drier conditions during high PDO phases" P 2 Line 14. "A reduced sea-ice cover during summer is observed in the region during PDO- (NPI+)" I do not understand. PDO is negatively correlated to precipitation but positively correlated to sea-ice cover during summer? Could you please simplify and clarify the description of the processes?

P 3 Line 16. It is really not clear what these correlations indicate, where we can see these correlations and why you speak of this in the introduction.

P 3 L 20. this paragraph is material and not introduction.

P 7 L 5. "When a 5-year running mean is applied on the series, the coherence between both records is much stronger (Fig. 4b: r = -0.39)." This is probably not true. You must take into account the reduction of degrees of freedom due to smoothing. Same

comment for the line just after.

P 8 L 18. "Hence the two modes, during AO+ and NPI+, might constructively interfere to strengthen northerly winds over the Arctic," I do not know if they "constructively interfere" or if they share in part the same signal.

P 10 L 8. "suggesting some potential for decadal-scale climate prediction." Could you please further elaborate?

Technical corrections

P 4 L 14. the sentence must be replaced with "a dataset that provides robust observations"

P 4 L 15. "The PDO as defined in (1997)" By whom?

P 4 L 17. "A second PDO index, based on the Extended Reconstructed Sea Surface Temperature (ERSSTv4) dataset . . . was constructed by regressing the ERSSTv4 anomalies against the Mantua PDO index using the period of overlap, resulting in a PDO regression map for North Pacific ERSST anomalies." Sentence to be reworded.

P 5 L 13. Dee et al 2011. Reference not well cited.

P 8 L 14. "It has been shown that PDO and Arctic Oscillation (AO) when both are in a positive increase summer precipitation in regions of Alaska (L'Heureux et al., 2005)." Something wrong in the sentence?

P 8 L 17. "albeit slightly less significant results" ???

Figure 1. c shows time series and not correlations.

Figure 2. c shows time series and not correlations.

Figure 3. I do not understand from the legend if one time series was shifted by 2 years.

---

## Author Comment (AC1) · 22 Feb 2017

Firstly, we would like to thank the first referee for the constructive comments. These will increase the quality of the manuscript.

General comments

1. We agree that additional discussion of dating accuracy should be included in the main text. Furthermore, it would make it easier for the reader to have this discussion in the main text body instead of in the supplemental material. This will be placed in section 2.3. Thanks for this comment. The lower frequency climatic signal in the varve record is seen when a 25-year low-pass filter is applied to both our record and the

millennial MacDonald and Case (2005) PDO (Supplemental Figure S5). This could also be inserted in the main text.

2. While we find that this is a very interesting point, it is not the scope of this paper to make a link between the AO and PDO, but we agree that this relationship should be more deeply analysed in modern and instrumental climate studies. We hope that our work will attract the attention of researcher working on that topic. Nevertheless, we think the PDO (NPI) and the AO partly share the same signal since they are correlated over the past 100 years (r = 0.45). Therefore, we added some text (highlighted in yellow) in the section mentioning the potential influence of the AO and we added references that further explain the potential relationships between the AO and PDO (NPI).

3. We totally agree with this comment and since our record looks quite convincing when correlated to the PC1 of the PDOs used here, this hypothesis makes a lot of sense; We will thus add this comment to the discussion.

4. There is one other varve record located nearby, Nicolay Lake (Lamoureux 2000), Cornwall island, located 470 km northeast of CBEL. It is negatively correlated to the PC1 of the PDOs at the annual scale (R = -0.21, p = 0.003) and using a 5 year-running mean it only increases slightly (R = -0.28). This record is shorterÂă: 500 years. Moroever, compared to Cape Bounty, we have a less comprehensive knowledge of the processes occuring within Nicolay Lake's watershed. Nicolay Lake system seems to working differently, and has so far been shown to be mainly sensitive to rainfall events. Therefore, we think it is not appropriate to compare the two records in this paper, although we are planning on going back to Nicolay Lake to apply the new techniques (XRF, Grain-size from thin-sections) that have been developed since Nicolay Lake has been investigated in the 90s.

5. Ok, thanks.

6. An english native has passed through it now.

Minor points

- References in the abstract were removed. Thank you.

- We reworded the negative correlation for the past century (instrumental) and for the last 7 centuries (reconstructed-PDO).

- ENSO: added reference by Rob Allan, thank you.

- Cape Bounty East Lake is 320 km southeast of Mould Bay: added in the methods, section 2.2.

- Show varve site on map: OK.

- Mantua is now added before (1997).

- Ok, a discussion on errors in dating is added for this 2.3 section

- MSLP is now being used instead of mslp, thank you.

- erosive bed: we reformulated this sentence.

- Factor loadings of the PC1 are 0.58 (D'Arrigo et al. 2001), 0.68 (Gedalof and Smith 2001) and 0.65 (MacDonald and Case 2005). This is now included in the main text.

- The 18 and 28 year lag are indeed large offsets. Unfortunately, in varves studies from the Arctic (and probably in other environment), it is clear that missing and/or adding extra varves might occur (Ojala et al. 2012; full citation found in the text). Also, in arctic areas, the hypothesis that the upper part of a lake was ice-frozen for years can not be ruled out. If this would occur, no clastic input would reach the lake bottom, making offsets unavoidable. The huge lack of similar high-resolution records in this region impedes a more reliable chronological control. Nevertheless, as explained in the text, all of the present-day teleconnection using instrumental and reanalysis correlations support our assertion that this region is influenced by these climate modes.

---

## Author Comment (AC2) · 22 Feb 2017

We would like to thank the referee 2 for the detailed comments.

General comments

We tried to clarify the text in order to better explain that we make a comparison of a varved record with PDO observations/reconstructions, AND (and not or) that this observation leads us to suggest that PDO had an influence over the western Arctic over the last centuries.

We feel that the abstract is correct but we rewrote parts of it (highlighted in yellow).

[Figure]

Specific comments

- Introduction: We now provide more information about the objectives.

- Section 2.2: this sections has been changed; the last paragraph of the introduction was transferred in a new section 2.1 (highlighted in yellow).

- Section 3.3: Spectral analysis: Do you think spectral analysis can also be influenced by the origin of the data (tree-ring, varved records) AR: Yes it can. Since these are annual archives they use to have significant spectra at higher frequencies range (1-5 year cycles), that might be confused with white noise. However, for the longer variability (>10 years), should not have such an impact.

Using the raw data and applying spectral analysis, we get similar results (see figure below): the decadal (19-26) and multidecadal (67-87) signals are also observed.

- Comments: suggesting drier conditions during high PDO phases, and vice-versa. AR: we agree that this sentence is hard to understand and we have re-write this sentence (highlighted in yellow).

- P.3 L 16 Where we can see these correlations: Figure 2. Why you speak of this in the introduction: This has been removed.

- P3 L20: This has been moved to the material section. It is indeed better into the material, thanks for this suggestion.

- P7 L5: 5 year-running mean: We agree that we must take into account the degrees of freedom. However, all the annual PDOs, including the PC1, are correlated significantly without any smoothing. In that respect, we applied a 5 year running mean because it makes sense for comparison purposes since the PDO is a decadal to multidecadal mode of variability.

- P8 L18. We totally agree with this comment that the AO and NPI might share the same signal. We added this in our text.

- P10 L8. suggesting some potential for decadal-scle climate prediction AR: We have changed for: Given the oscillatory nature of the PDO, there is some potential for decadal-scale climate prediction using the kind of record shown here. In that sense, more high-resolution records with longer timescales from this region could be beneficial for future PDO projection.

- P4 L15: In Mantua. This has been added, thanks.

- P4 L17: sentence rewritten. Thank you.

- P5 L13: Now in the reference: Dee et al. Thanks.

- P8 L 14: This sentence has been reworked and highlighted in yellow.

- P8 L17: This has been removed.

- Figure 1 and 2 c: thanks we changed them correctly.

- Figure 3: Yes, the CBEL was shifted 1 year, but there is no lag compared to the NPI (Figure S2). This is now placed in the text.
* * *
[Figure]

**Fig. 1.**

---

## Author Response (AR2)

Dear Dr. Zorita, we would like to thank you for giving us the opportunity to respond to the referees and make the corresponding corrections.

Moreover, we considered your new comments :

1) Reviewer #2 had some concerns regarding the abstract. I still feel that the abstract may mislead the reader, especially the opening sentences related to the ice-albedo feedback. The study does touch on this feedback but only marginally. Actually, the word albedo appears only in the abstract, which indicates that this is not a central topic of the study. I think it may mislead the reader along a wrong direction. Also, your study is not related to external forcings or to separating the influence of external and internal variability. It is rather a study on the link between the varved record with the PDO on long-time scales and an analysis of this record

We removed the opening sentence, and also the external forcing from the abstract. We now begin with : Understanding how internal climate variability influences arctic regions is required to better forecast future global climate variations.
Thank you.

2) The statistical significance of the interannual correlations is always indicates, but not of the correlations after time filtering. I am aware that this significance is harder to estimate due to to the built-in auto-correlation of the series. However, quite often the claim is made in the manuscript that correlations increase after time filtering. It would be very useful if the significance levels could be indicated also in these cases. Thy can be estimated by several approximate methods, either by calculating an equivalent number of degrees of freedom, or by Monte Carlo Simulations

We now performed a nonparametric stationary bootstrap, using 1000 iterations, to adress the significance of the raw and the filtered time series (Mudelsee, 2010; citaiton in main text). We added a table (Table 1) containing p-values and correlation coefficients. All of the correlations increase substantially after time filtering. Box-plots of the p-values can be seen below.
 Thanks.

[Figure]

p-values for the correlations between (raw and filtered) reconstructed PDOs and
CBEL VT using a nonparametric stationary bootstrap (1000 iterations). Red line is the
95% confidence levels.
3) I would suggest to careful reconsider the very last sentence of the
manuscript (line 649). Future precipitation changes will be driven by
several factors. This sentence refers to changes in regional
atmospheric dynamics and possibly local evaporation from land, but
one very important factor will be the increase in atmospheric humidity
due to rising global temperatures and changes in the transport of this
humidity by the large-scale circulation. So it is difficult to estimate the
total change in precipitation - actually the agreement shown by
climate models regarding future precipitation changes is generally
rather poor. But I also failed to see the significance of this sentence in
the context of your study. If one is interested in estimating future
precipitation changes in this area, one can just look directly at climate
simulations, the very same that also predict changes in the PDO or in
MSLP.
The sensences mentioning changes in precipitation were removed. We added these
highlighted sentences in the conclusion.
==An important finding from this study is the reduced sea-ice cover observed==

==during PDO-, which is in agreement with simulations made from Screen and Francis==

==(2016).== The PDO – western Canadian Arctic relationship has persisted at least for the past ~700 years as revealed by the strong coherence between the CBEL varve record and multiple PDO reconstructions. ==Given the oscillatory nature of the PDO, there is==
==some potential for improved constraint over decadal-scale climate prediction using the==
==kind of sedimentary record shown here, which in turn could give insights into future sea-==
==ice variability. In that sense, more high-resolution records with longer timescales from==
==this region could be beneficial for future PDO projection.==

Response to the referees
(writings in blue are comments; red are changes made in the text; black are referees
comments)
Firstly, we would like to thank the two referees for the constructive comments. These
will increase the quality of the manuscript.
General comments :

1. I would like to have seen more discussion of dating accuracy in the main text,
particularly given the discussion of sizeable errors of 18, 28 years in lags with the
paleoclimatic comparisons. Also on how well the varves reflect lower frequency climatic
information.

1. We agree that additional discussion of dating accuracy should be included in the
main text. Furthermore, it would make it easier for the reader to have this
discussion in the main text body instead of in the supplemental material. This is
now placed in section 2.3. Thanks for this comment.
The lower frequency climatic signal in the varve record is seen when a 25-year
low-pass filter is applied to both our record and the millennial MacDonald and
Case (2005) PDO (Supplemental Figure S5). We added a sentence on this:
(here p. 20, line 544) The comparison between CBEL and the PDO from
MacDonald and Case (2005) depicts a strong co-variability at longer-frequencies
(25-year low-pass filter applied on those time-series : r = -0.69, supplementary
Figure S5), suggesting a link between the lower frequency component of the
PDO and the regional climate of the western Canadian Arctic.

2. Perhaps some discussion of whether the PDO is the most significant influence to discuss here, rather than the Arctic Oscillation.

While we find that this is a very interesting point, it is not the scope of this paper to make a link between the AO and PDO, but we agree that this relationship should be more deeply analysed in modern and instrumental climate studies. We hope that our work will attract the attention of researcher working on that topic. Nevertheless, we think the PDO (NPI) and the AO partly share the same signal since they are correlated over the past 100 years (r = 0.45). Therefore, we added some text (highlighted in yellow) in the section mentioning the potential influence of the AO and we added references that further explain the potential relationships between the AO and PDO (NPI).

(p.22, line 582) It has similarly been shown that PDO and Arctic Oscillation (AO) are useful determinants of precipitation characteristics during summer season in regions of Alaska (L'Heureux et al., 2004) and positive AO index has been linked to reduced sea-ice extent and increased atmospheric heat transport into the Arctic (Rigor et al., 2002; Zhang et al., 2003). The correlation between the AO and the meridional windstress anomalies (Fig. 6b) yields very similar pattern as the NPI (Fig. 6a). This is not too surprising, since these two climate indices are significantly correlated during SON (1900-2015: r = 0.45, p < 0.0001). Hence the two modes which may share in part the same signal might constructively interfere to strengthen northerly winds over the Arctic during AO+ and NPI+, converging with southerly moisture-laden winds from the North Pacific over the western Canadian Arctic, thereby favoring precipitation in the region during autumn.

3. The tree-ring reconstructions of the PDO vary in part because of the different geographical representation of the sites used in each case. Another PDO reconstruction based on tree rings is that of Biondi. As a result I believe that they are best interpreted as reflections of the PDO at their given study sites

We totally agree with this comment and since our record looks quite convincing when correlated to the PC1 of the PDOs used here, this hypothesis makes a lot of sense; We will thus add this comment to the discussion.

(p. 19, line 519) These reconstructed PDOs are probably best interpreted as reflections of the PDO at their given study sites, explaining the lack of co-variability during certain periods.

4. Are there perhaps other varve/paleo records in the vicinity of the varve site that might be more appropriate for comparison?

| | |
|---|---|
| 143 | There is one other varve record located nearby, Nicolay Lake (Lamoureux 2000), |
| 144 | Cornwall island, located 470 km northeast of CBEL. It is negatively correlated to |
| 145 | the PC1 of the PDOs at the annual scale (R = -0.21, p = 0.003) and using a 5 year- |
| 146 | running mean it only increases slightly (R = -0.28). This record is shorter : 500 |
| 147 | years. Moroever, compared to Cape Bounty, we have a less comprehensive |
| 148 | knowledge of the processes occuring within Nicolay Lake's watershed. Nicolay |
| 149 | Lake system seems to working differently, and has so far been shown to be |
| 150 | mainly sensitive to rainfall events. Therefore, we think it is not appropriate to |
| 151 | compare the two records in this paper, although we are planning on going back |
| 152 | to Nicolay Lake to apply the new techniques (XRF, Grain-size from thin-sections) |
| 153 | that have been developed since Nicolay Lake has been investigated in the 90s. |
| | |
| 154 | 5. Good to note the issue of seasonality – trees reflect conditions during different |
| 155 | seasons than the varves..also that the dating is more precise |
| | |
| 156 | Ok, thanks. |
| 157 | |
| | |
| 158 | 6. Some (mostly light) editing of English would benefit the manuscript |
| | |
| 159 | We have edited the english of the whole manuscript. Thank you. |
| 160 | |
| 161 | |
| 162 | Minor points : |
| 163 | |
| | |
| 164 | Abstract: References ok in abstract? |
| | |
| 165 | - References in the abstract were removed.  Thank you. |
| | |
| 166 | Line 9, reword to note that nega- tively correlated with instrumental for past century, |
| 167 | recons over past centuries to 700 years.. |
| | |
| 168 | - Ok. We reworded the negative correlation for the past century (instrumental) |
| 169 | and for the last 7 centuries (reconstructed-PDO). |
| | |
| 170 | (p. 11, line 338) This paper investigates an annually laminated (varved) record from |
| 171 | the western Canadian Arctic and finds that the varves are negatively correlated with |
| 172 | both the instrumental Pacific Decadal Oscillation (PDO) during the past century and |
| 173 | also with reconstructed PDO over the past 700 years |
| | |
| 174 | P. 3 line 7 ENSO references by Rob Allan, Hadley Centre relevant here |
| | |
| 175 | - ENSO : added reference by Rob Allan, thank you. |

Allan, R., Lindesay, J., and Parker, D.: El Niño southern oscillation & climatic variability, CSIRO publishing, 1996, 406 pages.

p. 3 line 20 show varve site on map.

Done. Thanks.

How far from Mould Bay?

- Cape Bounty East Lake is 320 km southeast of Mould Bay : added in the methods, section 2.2. Thank you.

p. 4 line 16 Mantua, 1997

- Mantua is now added before (1997). Thanks.

p. 5 first paragraph: good to discuss errors in dating a bit here in main text..

- Ok, a discussion on errors in dating is added for this 2.3 section p. 6 line 2: MSLP not mslp

- MSLP is now being used instead of mslp, thank you.

p. 6 line 22: inference not clear re erosive bed and how this relates to first part of sentence

- erosive bed : we reformulated this sentence :
(p. 19, line 515) For the time interval 1300-1900 CE, a single 1.34 cm thin erosive bed is evident in the sedimentary record (Supplemental Text 1, Supplementary Fig. S4), making the comparison of the CBEL varve thickness (VT) with other paleo-PDOs acceptable p. 6 line 26: what are the loadings of the three recons in PC1?

- Factor loadings of the PC1 are 0.58 (D'Arrigo et al. 2001), 0.68 (Gedalof and Smith 2001) and 0.65 (MacDonald and Case 2005). This is now included in the main text; highlighted in yellow.

p. 7 line 1 18 year lag: this seems like a rather large offset. 7 line 10: ditto 28 year lag..

-    The 18 and 28 year lag are indeed large offsets. Unfortunately, in varves studies
from the Arctic (and probably in other environment), it is clear that missing
and/or adding extra varves might occur (Ojala et al. 2012; full citation found in
the text).
Also, in arctic areas, the hypothesis that the upper part of a lake was ice-frozen for
years can not be ruled out. If this would occur, no clastic input would reach the lake
bottom, making offsets unavoidable. The huge lack of similar high-resolution records
in this region impedes a more reliable chronological control. Nevertheless, as
explained in the text, all of the present-day teleconnection using instrumental and
reanalysis correlations support our assertion that this region is influenced by these
climate modes.

Reviewer 2
We would like to thank the referee 2 for the detailed comments.
General comments :

I think this is a potentially nice study on the influence of PDO on Western Canadian
Arctic and on the mechanisms relating PDO and a varved record. However the main
concern with the paper is that the authors do not clearly state their objectives and the
links between the paper sections. At the end of the introduction we do not know if the
paper is mostly a comparison between a varved record and PDO obser-
vations/reconstructions or if the authors want to study the PDO influence over the last
century with correlations.

We tried to clarify the text in order to better explain that we make a comparison of a
varved record with PDO observations/reconstructions, AND (and not or) that this
observation leads us to suggest that PDO had an influence over the Western Arctic over
the last centuries.

(p.13, line 384) Here, instrumental and reanalysis meteorological data combined with
sedimentological evidence highlight that this remote region is influenced by the PDO.

The main objectives are now displayed more thoroughly in the introduction. Thank you.
(p. 12, line 373) In the recent years, several varved records have been established in the
Arctic (at Cape Bounty: Cuven et al. (2011); Lapointe et al. (2012), at South Sawtooth
Lake: Francus et al. (2008), at Lake C2: Douglas et al. (1996), at Murray Lake: Besonen et
al. (2008) and Lower Murray Lake: Cook et al. (2009)) in order to investigate past climate variations. Amongst them, the Cape Bounty record is most probably the best documented because it has been supported by climate, hydrological, and limnological research at the Cape Bounty Arctic Watershed Observatory since 2003. The annual nature of this sedimentary record, its duration (700 years), and the above-average quality of its chronology opens the opportunity to investigate (1) correlations with instrumental records, (2) cyclities of this record by time-series analysis, (3) teleconnections with major climate indices, and (4) the long-term influence of the climate mode of variability on the western Canadian Arctic.

Specific comments

The abstract must be reworded. This is mostly a comparison between a varved record and PDO observations/reconstructions (P 2 L 6. "Here, sedimentological evidence from an annually laminated (varved) record highlights that North Pacific climate vari- ability has been a persistent regulator of the regional climate in the western Cana- dian Arctic."). The conclusion of the abstract (P 2 line 15-20) says nothing on the results/implications of THIS paper. (PS Now that I have finished to read the paper I have partially changed my mind on this comment, however I see that the problem is that you do not clearly state your objectives and the methods you apply to reach them)

We feel that the abstract is correct but we rewrote parts of it (highlighted in yellow).

Section 2.2. You must describe here your data. Not at the end of the introduction which is the place for objectives.

- Section 2.2 : this sections has been changed; the last paragraph of the introduction was transferred in a new section 2.1 (highlighted in yellow). Thanks.

Section 3.3. Do you think that the spectral analysis can also be influenced by the origin of the data (tree-rings, varved records) and not only by the modes? For example, you use a box-cox transformation to stabilize variance in your time series. What do you get in terms of spectral analysis if this transformation is not applied?

- Yes it can. Since these are annual archives they use to have significant spectra at higher frequencies range (1-5 year cycles), that might be confused with white noise. However, for the longer variability (>10 years), should not have such an impact.

Using the raw data and applying spectral analysis, we get similar results (see below) :
the decadal (19-26) and multidecadal (67-87) signals are also observed.

[Figure]

Spectral analysis using the raw varve thickness series.

P 2 Line 10. "suggesting drier conditions during high PDO phases" P 2 Line 14. "A re-
duced sea-ice cover during summer is observed in the region during PDO- (NPI+)" I do
not understand. PDO is negatively correlated to precipitation but positively correlated
to sea-ice cover during summer? Could you please simplify and clarify the description of
the processes?

- we agree that this sentence is hard to understand and we have re-write this sentence
(highlighted in yellow).
(p. 11, line 348) Reduced sea-ice cover during summer-autumn is observed in the region
during PDO- (NPI+) and is associated with low-level southerly winds that originate from
the northernmost Pacific across the Bering Strait and can reach as far as the western
Canadian Arctic.

P 3 Line 16. It is really not clear what these correlations indicate, where we can see
these correlations and why you speak of this in the introduction.

-     Where we can see these correlations : Figure 2. Why you speak of this in the
 introduction : This has been removed from the introduction.

P 3 L 20. this paragraph is material and not introduction.

-     This has been moved to the material section. It is indeed better into the
 material, thanks for this suggestion.

P 7 L 5. "When a 5-year running mean is applied on the series, the coherence between
both records is much stronger (Fig. 4b: r = -0.39)." This is probably not true. You must
take into account the reduction of degrees of freedom due to smoothing. Same
comment for the line just after.

-     P7 L5 : 5 year-running mean : We agree that we must take into account the
 degrees of freedom. However, all the annual PDOs, including the PC1, are
 correlated significantly without any smoothing. In that respect, we applied a 5
 year running mean because it makes sense for comparison purposes since the
 PDO is a decadal to multidecadal mode of variability.

P 8 L 18. "Hence the two modes, during AO+ and NPI+, might constructively interfere to
strengthen northerly winds over the Arctic," I do not know if they "constructively
interfere" or if they share in part the same signal.

-     P8 L18. We totally agree with this comment that the AO and NPI might share the
 same signal. We added this in our text as suggested by referee 1.

P 10 L 8. "suggesting some potential for decadal-scale climate prediction." Could you
please further elaborate?

We added this sentence to make it clearer (p. 25, line 645) Given the oscillatory nature of the PDO, there is some potential for
improved constraint over decadal-scale climate prediction using the kind of sedimentary
record shown here. In that sense, more high-resolution records with longer timescales
from this region could be beneficial for future PDO projection.

Technical corrections

P 4 L 14. the sentence must be replaced with "a dataset that provides robust
observations"

-    Done. Thank you.

P 4 L 15. "The PDO as defined in (1997)" By whom?

-    P4 L15 : In Mantua. This has been added, thanks.

P 4 L 17. "A second PDO index, based on the Extended Reconstructed Sea Sur- face
Temperature (ERSSTv4) dataset . . . was constructed by regressing the ERSSTv4
anomalies against the Mantua PDO index using the period of overlap, resulting in a PDO
regression map for North Pacific ERSST anomalies." Sentence to be reworded.

-    P4 L17 : sentence rewritten.
(p.420, line 14) A second PDO index was constructed by regressing the Extended
Reconstructed Sea Surface Temperature (ERSSTv4) (Huang et al., 2015) temperature
anomalies against the Mantua PDO index during the period of overlap. This resulted
in a PDO regression map for North Pacific ERSST anomalies. This index closely
resembles the Mantua PDO index.

P 5 L 13. Dee et al 2011. Reference not well cited.

Ok done. Thank you.

Dee, D., et al.: The ERA-Interim reanalysis: Configuration and performance of the data
assimilation system, Q. J. R. Meteorol. Soc. , 137, 553-597, doi : 10.1002/qj.828, 2011.

P 8 L 14. "It has been shown that PDO and Arctic Oscillation (AO) when both are in a
positive increase summer precipitation in regions of Alaska (L'Heureux et al., 2005)."
Something wrong in the sentence?

-    P8 L 14. This sentence has been reworked and highlighted in yellow.
(p. 22, line 585) It has similarly been shown that PDO and Arctic Oscillation (AO) are
useful determinants of precipitation characteristics during summer season in regions of

Alaska (L'Heureux et al., 2004) and positive AO index has been linked to reduced sea-ice extent and increased atmospheric heat transport into the Arctic (Rigor et al., 2002; Zhang et al., 2003). The correlation between the AO and the meridional windstress anomalies (Fig. 6b) yields very similar pattern as the NPI (Fig. 6a). This is not too surprising, since these two climate indices are significantly correlated during SON (1900-2015: r = 0.45, p < 0.0001). Hence the two modes which may share in part the same signal might constructively interfere to strengthen northerly winds over the Arctic during AO+ and NPI+, converging with southerly moisture-laden winds from the North Pacific over the western Canadian Arctic, thereby favoring precipitation in the region during autumn.

P 8 L 17. "albeit slightly less significant results" ???

- P8 L17. This has been removed. Thanks.

Figure 1. c shows time series and not correlations. Figure 2. c shows time series and not correlations.

- Figure 1 and 2 c : thanks we changed them correctly.

Figure 3. I do not understand from the legend if one time series was shifted by 2 years.

- Figure 3 : Yes, the CBEL was shifted 1 year (not 2 years), but there is no lag compared to the NPI (Figure S2). This is now in the text.

[revised manuscript text omitted]

(a) Correlation between (JAS) PDO and mean sea-ice cover (b) Correlation between (JAS) PDO and mean sea level pressure (c) r = -0.47, p < 0.0001

(d) Correlation between (JFM) PDO and mean sea level pressure

**Figure 1.** PDO modulation of Western Canadian Arctic climate. (a), Correlation between PDO (Huang et al., 2015) and sea-ice anomalies from ERA-Interim (Dee et al., 2011) for July-September during 1979-2016. (b), as in a) but for mean sea level pressure from ERA-Interim (Dee et al., 2011). (c), Comparison between the time series of rainfall at Mould Bay and PDO during July-September. (d), as in b) but for January-March (JFM). Black and green asterisks denote Cape Bounty and Mould Bay weather  station, respectively. Note that Mould Bay weather station stopped operating in 1996.

[Figure]

**Figure 2.** North Pacific Index (NPI) and precipitation during September-November. (a), Correlation between NPI (Trenberth and Hurrell, 1994) and mean sea level pressure from 1979-2015. (b), Same as (a), but for precipitation anomalies (Dee et al., 2011) correlated with NPI index. Black and green asterisks denote Cape Bounty and Mould Bay weather station, respectively. (c) Comparison between the time series at Mould Bay snow depth and NPI during September-November (Trenberth and Hurrell, 1994). Note that Mould Bay weather station stopped operating in 1996 CE.

[Figure]

**Figure 3.** Instrumental PDO (NOAA) compared with grain size at Cape Bounty
East Lake from 1900-2000. (best correlation is achieved when CBEL lags PDO by 1 year).
Bold lines are 10-year low-pass filtered. Seven years were eroded by a large turbidite
dated to 1971 CE (Lapointe et al. 2012).

[Figure]

**Figure 4**. a), Comparison between normalized Cape Bounty East Lake varve
thickness and normalized PDO from MacDonald and Case (2005) (VT is shifted 18 years
earlier). b), Same as A) but for the PDO from Gedalof and Smith (2001). c), Same as a)
and b) but using the PDO from D'Arrigo et al. (2001). d), Same as a), b) and c) but using
the PC1 extracted from PCA analysis of the three PDOs. Time series are filtered by a 5-
year running-mean.

(a)

[Figure]

(b)

[Figure]

**Figure 6**. a), Spectral analysis of the varve thickness series. After Schulz and Mudelsee (Schulz and Mudelsee, 2002). b), Wavelet analysis: black boundaries show the 95% confidence level based on a red noise process. White shading represents the cone of influence where edge effects might be important.

[Figure]

**Figure 7.** a), Correlation between NPI (Trenberth and Hurrell, 1994) and
meridional windstress anomalies from 1950-2015. b), Same as a), but for correlation
with Arctic Oscillation index (derived from NCEP/CPC). Note that the Era-Interim yields
similar result (not shown). Black and green asterisks denote Cape Bounty and Mould Bay
weather station, respectively.

[Figure]

**Figure 8**. PDO modulation of winds and sea surface temperature in the Pacific.
From Zhang and Delworth (Zhang and Delworth, 2015). Regression of SST (°C) and
surface wind stress (N m$^2$) against the PDO index. Note the northward direction of the
wind stress in the central northern part of the Pacific during the negative phase of the
PDO (b). Winds from the Siberian shelf have an eastward direction and reach Melville
Island during negative PDO. Reproduced with permission from the American
Meteorological Society (AMS).